# Beam Quality Factor of Partially Coherent Airy Beam in Non-Kolmogorov Turbulence

**Wei Wen [1,2,*], Xianwu Mi [2] and Sirui Chen [1]**

[1] College of Mathematics and Physics Science, Hunan University of Arts and Science, Changde 415000, China

[2] Key Laboratory of Intelligent Control Technology for Wuling-Mountain Ecological Agriculture in Hunan Province, HuaiHua University, Huaihua 418000, China

[*] Correspondence: huaswenwei@126.com; Tel.: +86-188-9070-7239

**Abstract:** A universal formula of the beam quality factor for a partially coherent Airy (PCA) beam in non-Kolmogorov turbulence has been investigated. Results of numerical simulation display that expanding the internal scales of non-Kolmogorov turbulence is good for decreasing the beam quality factor of a PCA beam. Another benefit of reducing the beam quality factor is decreasing the generalized structure constant and the outer scales of non-Kolmogorov turbulence. Similarly, the quality factor decreases with increasing transverse scale of a PCA beam. In the meantime, decreasing a laser beam's coherence length also leads to the quality factor's degeneration. What is more, the beam quality factor reaches the maximum value as the generalized exponent parameter of the turbulence is 3.1.

**Keywords:** Airy beam; beam quality factor; non-Kolmogorov turbulence

## 1. Introduction

The proper noun "LASER" is an oscillator that operates at very high frequencies. Since 1960, the application of laser has developed rapidly. Application of laser technology always plays an important role in modern society due to the widespread application prospect for the communications industry, materials processing, medical imaging techniques, laser tweezers and data storage [1,2].

Since the appearance of the laser, people have begun to study the quality of the laser beam. In modern laser research and development, the laser beam quality is very important, and a good quality factor of a laser beam means near diffraction is limited [3,4]. There are many definitions of laser beam quality [3,5,6]. Among these definitions, the beam quality factor is widely used [7,8]. In fact, the $M^2$ factor (beam quality factor) is nearly always specified when a new laser system is reported [9–16]. It is ascertained that the $M^2$ factor remains unvarying when a laser beam overgoes all kinds of focusing optical equipment or free space [17–19]. However, the oscillations of the turbulence have an influence on the quality of a laser beam directly [20–26].

Historically, the atmosphere turbulence causes effects on laser beams that have been very interesting to scientists for a long time [27–29]. Optical turbulence is the oscillations in the refractive index of air following as an effect of small temperature fluctuations [29]. It is generally accepted that the optical turbulence is statistically average and optically isotropic in three-dimensional space (Kolmogorov theory). However, recent atmospheric experiment results are different from the Kolmogorov theory [30–33]. Optical turbulence displaying non-Kolmogorov characteristics is regarded as non-Kolmogorov turbulence. According to the theory of non-Kolmogorov, the beam quality factor of completely coherent beams [34,35], Gaussian Schell-model beams [36–40] and lasers with controllable coherence [41–43] have been probed. However, the beams that have been mentioned above tend to propagate along a straight path.

The accelerating Airy beam, which was observed by Siviloglou and Christodoulides, is a novel beam that can bend itself along a curved path on the plane of propagation [44–46]. Due

to the fact that the curved light field permits a huge degree of safety for optical communication, the transmission characteristics of a curved Airy beam passing through turbulent atmosphere [47–52] were discussed. In 2011, the $M^2$ for Airy beams through optical free space was investigated by Chen and Ying [53]. However, the mechanisms responsible for non-Kolmogorov statistics on the PCA beams' $M^2$ factor still need to be discussed.

In this paper, we consider the $M^2$ of PCA beams in non-Kolmogorov medium. A universal formula of the $M^2$ factor for PCA beams in non-Kolmogorov medium have been found. The elements that influence the $M^2$ factor pass through non-Kolmogorov turbulence are batted around by numerical discussion.

## 2. Formulation

On the initial plane, the Airy beam's expression is given by [44,45,47]

$$E(x, y, 0) = Ai(x/w_0) Ai(y/w_0) \exp[a(x + y)/w_0], \tag{1}$$

It should be noted that the $Ai(\cdot)$ is an integral expression with $Ai(s) = \frac{1}{2\pi} \int_{-\infty}^{\infty} e^{i(t^3/3+st)} dt$. The symbol a is decay factor. The $w_0$ is called transverse scale, which determines the range of the laser spot.

The cross-spectral density (CSD) function of a PCA beam at z = 0 is

$$W(\mathbf{r}_1, \mathbf{r}_2, 0) = E(\mathbf{r}_1, 0) E^*(\mathbf{r}_2, 0) \exp\left[-\frac{(\mathbf{r}_1 - \mathbf{r}_2)^2}{\sigma_g^2}\right], \tag{2}$$

where $\sigma_g$ is called the laser's coherence length.

Based on the transmission equation, the CSD function in non-Kolmogorov turbulence at the receiving plane is [20,29]

$$W(\mathbf{r}, \mathbf{r}_d; z) = \frac{4\pi^2}{k^2 z^2} \int_{-\infty}^{\infty} \int_{-\infty}^{\infty} W(\mathbf{R}, \mathbf{R}_d; 0) \exp\left[\frac{ik}{z}(\mathbf{r} - \mathbf{R})(\mathbf{r}_d - \mathbf{R}_d) - H(\mathbf{r}_d, \mathbf{R}_d; z)\right] d^2\mathbf{R} d^2\mathbf{R}_d, \tag{3}$$

with

$$\mathbf{r} = (\mathbf{r}_1 + \mathbf{r}_2)/2, \quad \mathbf{r}_d = \mathbf{r}_1 - \mathbf{r}_2, \quad \mathbf{R} = (\mathbf{R}_1 + \mathbf{R}_2)/2, \quad \mathbf{R}_d = \mathbf{R}_1 - \mathbf{R}_2. \tag{4}$$

The notation $W(\mathbf{R}, \mathbf{R}_d; 0)$ is equivalent to the left of Equation (2). The notation $W(\mathbf{r}, \mathbf{r}_d; z)$ is the CSD functions of a PCA beam at the output planes. The $k = 2\pi/\lambda$ is wave number, and $\lambda$ is wavelength.

In Equation (3), the sign $\exp[-H(\mathbf{r}_d, \mathbf{R}_d; z)]$ comes from non-Kolmogorov turbulence along a propagation length, and $H(\mathbf{r}_d, \mathbf{R}_d; z)$ can be written as [20,54]

$$H(\mathbf{r}_d, \mathbf{R}_d; z) = 4\pi^2 k^2 z \int_0^1 d\xi \int_0^{\infty} [1 - J_0(\kappa |\mathbf{R}_d \xi + (1 - \xi)\mathbf{r}_d|)] \Phi_n(\kappa) \kappa d\kappa. \tag{5}$$

In Equation (5), the notation $J_0$ is a Bessel function. $\Phi_n(\kappa)$ is the spectrum of non-Kolmogorov turbulence with $\kappa$ as spatial frequency.

As we know, the beam quality factor of a PCA beam in non-Kolmogorov turbulence can be analyzed using the Wigner distribution function (WDF). The WDF of a PCA beam is [21]

$$h(\mathbf{r}, \boldsymbol{\theta}; z) = \left(\frac{1}{\lambda}\right)^2 \int_{-\infty}^{\infty} \int_{-\infty}^{\infty} W(\mathbf{r}, \mathbf{r}_d; z) \exp(-ik\boldsymbol{\theta} \cdot \mathbf{r}_d) d^2\mathbf{r}_d, \tag{6}$$

where the mark is $\boldsymbol{\theta} = (\theta_x, \theta_y)$.

So, we can determine that the WDF is

$$
\begin{aligned}
h(\mathbf{r}, \boldsymbol{\theta}; z) \;=\;& \frac{w_0^2}{16\pi^3\lambda^2} \int_{-\infty}^{\infty} \int_{-\infty}^{\infty} \sqrt{\frac{1}{(2a+iw_0\kappa_{dx})(2a+iw_0\kappa_{dy})}} \\
&\times \exp\left\{ \frac{i}{12}\left[ (2ia - w_0\kappa_{dx})^3 + \left(2ia - w_0\kappa_{dy}\right)^3 \right] \right\} \\
&\times \exp\left\{ -\left[ \frac{1}{4(2a+iw_0\kappa_{dx})w_0^2} + \frac{1}{2\sigma_g^2} \right](x_d + z\kappa_{dx}/k)^2 \right\} \\
&\times \exp\left\{ -\left[ \frac{1}{4(2a+iw_0\kappa_{dy})w_0^2} + \frac{1}{2\sigma_g^2} \right]\left(y_d + z\kappa_{dy}/k\right)^2 \right\} \\
&\times \exp(i\mathbf{r}\cdot\boldsymbol{\kappa}_d - ik\boldsymbol{\theta}\cdot\mathbf{r}_d) \exp\left[-H\left(\mathbf{r}_d, \mathbf{r}_d + \tfrac{z}{k}\boldsymbol{\kappa}_d; z\right)\right] d^2\boldsymbol{\kappa}_d d^2\mathbf{r}_d.
\end{aligned}
\tag{7}
$$

As discussed above, one finds the moments of the PCA beam are

$$
\langle x^{n_1} y^{n_2} \theta_x^{m_1} \theta_y^{m_2} \rangle = \frac{1}{P} \int_{-\infty}^{\infty} \int_{-\infty}^{\infty} x^{n_1} y^{n_2} \theta_x^{m_1} \theta_y^{m_2} h(\mathbf{r}, \boldsymbol{\theta}; z) d^2\mathbf{r} d^2\boldsymbol{\theta},
\tag{8}
$$

where P is [47]

$$
P = \int_{-\infty}^{\infty} \int_{-\infty}^{\infty} h(\mathbf{r}, \boldsymbol{\theta}; z) d^2\mathbf{r} d^2\boldsymbol{\theta} = \frac{w_0^2}{8\pi a} \exp\left(4a^3/3\right).
\tag{9}
$$

On substituting Equation (7) into Equation (8), we see that

$$
\left\langle \mathbf{r}(z)^2 \right\rangle = \frac{8a^3 w_0^2 + w_0^2}{4a^2} + \frac{4aw_0^2 z^2 + \sigma_g^2 z^2}{2aw_0^2 k^2 \sigma_g^2} + \frac{4\pi^2 z^3 T}{3},
\tag{10}
$$

$$
\langle \mathbf{r}(z) \cdot \boldsymbol{\theta}(z) \rangle = \frac{4aw_0^2 z + \sigma_g^2 z}{2aw_0^2 k^2 \sigma_g^2} + 2\pi^2 z^2 T,
\tag{11}
$$

$$
\left\langle \boldsymbol{\theta}(z)^2 \right\rangle = \frac{4aw_0^2 + \sigma_g^2}{2aw_0^2 k^2 \sigma_g^2} + 4\pi^2 z T,
\tag{12}
$$

with

$$
T = \int_0^{\infty} \Phi_n(\kappa)\kappa^3 d\kappa,
\tag{13}
$$

which denotes the strength of turbulence. The definition of the $M^2$ factor, which is based on second moments, is [20]

$$
M^2(z) = k\left[\left\langle \mathbf{r}(z)^2 \right\rangle\left\langle \boldsymbol{\theta}(z)^2 \right\rangle - \langle \mathbf{r}(z) \cdot \boldsymbol{\theta}(z) \rangle^2\right]^{1/2}.
\tag{14}
$$

Using Equations (10)–(14), we obtain

$$
M^2(z) = \left\{ \frac{(8a^3 + 1)\left(\sigma_g^2 + 4aw_0^2\right)}{8a^3\sigma_g^2} + \frac{16\pi^4}{3\lambda^2}\left[ \frac{3(8a^3+1)Tz}{4a^2} + \left(\frac{\sigma_g^2 + 4aw_0^2}{2aw_0^2 k^2\sigma_g^2}\right)Tz^3 + \pi^2 T^2 z^4 \right] \right\}^{1/2},
\tag{15}
$$

As $T = 0$, the result of the equation degenerates to $\sqrt{(8a^3 + 1)\left(\sigma_g^2 + 4aw_0^2\right)/8a^3\sigma_g^2}$.

The classical theory of turbulence, which was developed by Kolmogorov, concerns random fluctuations in velocity. This concept was later applied to temperature fluctuations. Optical turbulence is mainly caused by small fluctuations in temperature. Temperature fluctuations, in turn, cause the refractive-index fluctuations. People used to think that the spatial power spectrum of Kolmogorov theory was the reality of a turbulent atmosphere. However, recent experimental data reveal that the mechanisms responsible for the Kolmogorov spectrum still need to be discussed. On the authority of previous research results, the T of the non-Kolmogorov spectrum is [41].

$$T = \frac{A(\alpha)\widetilde{C}_n^2}{2(\alpha-2)} \left\{ \exp\left(\frac{\kappa_0^2}{\kappa_m^2}\right) \kappa_m^{2-\alpha} \Gamma\left(2-\frac{\alpha}{2}, \frac{\kappa_0^2}{\kappa_m^2}\right) \left[(\alpha-2)\kappa_m^2 + 2\kappa_0^2\right] - 2\kappa_0^{4-\alpha} \right\} \quad (16)$$

with

$$A(\alpha) = \Gamma(\alpha-1)\cos(\alpha\pi/2)/\left(4\pi^2\right) \quad (17)$$

$$\kappa_0 = 2\pi/L_0 \quad (18)$$

$$c(\alpha) = \left[\frac{2\pi}{3}A(\alpha)\Gamma\left(\frac{5-\alpha}{2}\right)\right]^{1/(\alpha-5)} \quad (19)$$

$$\kappa_m = c(\alpha)/l_0 \quad (20)$$

In Equations (16)–(20), $\Gamma(\cdot)$ is the gamma function. It should be pointed out, in particular, that the $\alpha$ is the generalized exponent parameter with a range of 3–4 generally. $l_0$ is the inner scales and $L_0$ is outer scales of the atmosphere turbulence. In particular, it should be noted that $\widetilde{C}_n^2$ is the generalized structure constant.

Substituting Equation (16) into Equation (15), we can obtain the final formula of th e $M^2$ factor. So far, our study shows that the changes in the propagation distance, the generalized exponent parameter, the generalized structure constant and the inner and outer scales of non-Kolmogorov turbulence alter the beam quality factor of a PCA beam. Furthermore, it was discovered that the $M^2$ factor varies with the changes in the exponential decay factor, the scale factor and the coherence length of the laser beam.

### 3. Numerical Results and Analysis

In the following work, the $M^2$ factor is studied by numerically. In accordance with the reference [55], we use a relative beam quality factor of $M^2(z)/M^2(0)$ to perform numerical analysis.

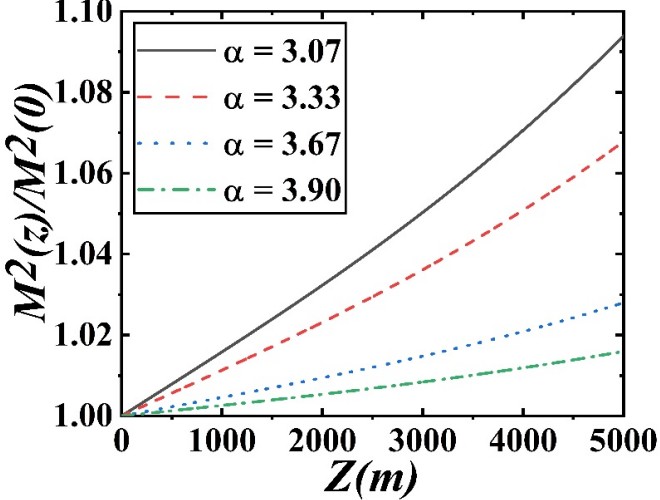

**Figure 1.** Dimensionless quantity $M^2(z)/M^2(0)$ of a partially coherent Airy beam in non-Kolmogorov turbulence versus propagation distance for different $\alpha$.

We show in Figure 1 the $M^2(z)/M^2(0)$ as a function of z for four generalized exponent parameters. The other parameters are $l_0 = 10\,\text{mm}$, $L_0 = 10\,\text{m}$, $\widetilde{C}_n^2 = 10^{-15}\,\text{m}^{3-\alpha}$, $a = 0.08$, $w_0 = 50\,\text{mm}$, $\sigma_g = 1\,\text{mm}$ and $\lambda = 1064\,\text{nm}$. From the graph, we see that the $M^2(z)/M^2(0)$ increases with the propagation length. The circumscription of the $M^2$ factor is the ratio of the spatial-beam-width of a certain beam to the spatial-beam-width of a fundamental mode Gaussian beam. So, the $M^2$ factor reflects the degree of quality departure from the ideal Gaussian beam. That is to say, the ideal Gaussian beam has the smallest $M^2$ factor (equal to 1). In this sense, the PCA beams' quality obviously degenerate with increasing z. Figure 1 shows also that the relative $M^2$ factor varies with $\alpha$. Clearly, the $M^2(z)/M^2(0)$ of

non-Kolmogorov turbulence (black, red and green lines in Figure 1) is distinct from that of Kolmogorov turbulence (blue line in Figure 1 and $\alpha = 11/3 \approx 3.67$).

The main difference between the above two kinds of turbulence is that there is a generalized parameter $\alpha$ in non-Kolmogorov turbulence. To learn more about the effect of the generalized exponent parameters $\alpha$ on the beam quality, we exhibit in Figure 2 the $M^2(z)/M^2(0)$ as a function of $\alpha$ for three propagation distances by numerical calculation. The other parameters in Figure 2 are the same as those in Figure 1. Obviously, the variation of the relative beam quality factor is non-monotonic. These results suggest that the poorest $M^2(z)/M^2(0)$ is achieved for $\alpha \approx 3.11$. In the region $3 < \alpha < 3.11$, the $M^2(z)/M^2(0)$ increases with increasing $\alpha$. However, the $M^2(z)/M^2(0)$ decreases with an increasing $\alpha$ in the range $3.11 < \alpha < 4$. In fact, the quantity $T(\alpha)$ represents the non-Kolmogorov turbulence's intensity in Equation (16). In the region $3 < \alpha < 3.11$, $T(\alpha)$ increases as $\alpha$ increases, while $T(\alpha)$ decreases as $\alpha$ increases in the region $3.11 < \alpha < 4$. The strongest turbulence is achieved at $\alpha \approx 3.11$, determined by $\partial T(\alpha)/\partial \alpha = 0$.

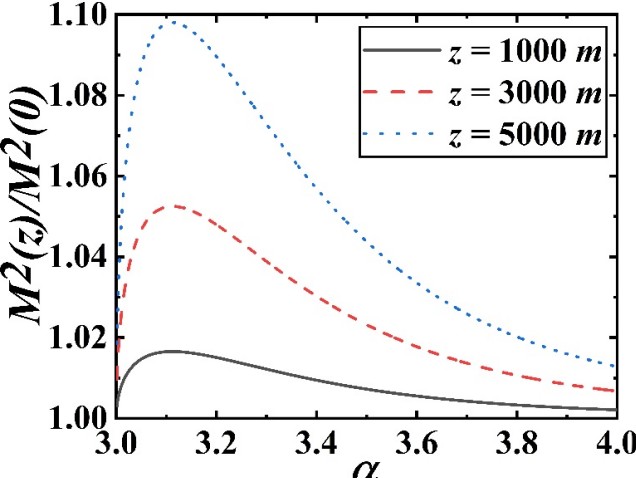

**Figure 2.** Dimensionless quantity $M^2(z)/M^2(0)$ of a partially coherent Airy beam in non-Kolmogorov turbulence versus the generalized exponent parameter for different $z$.

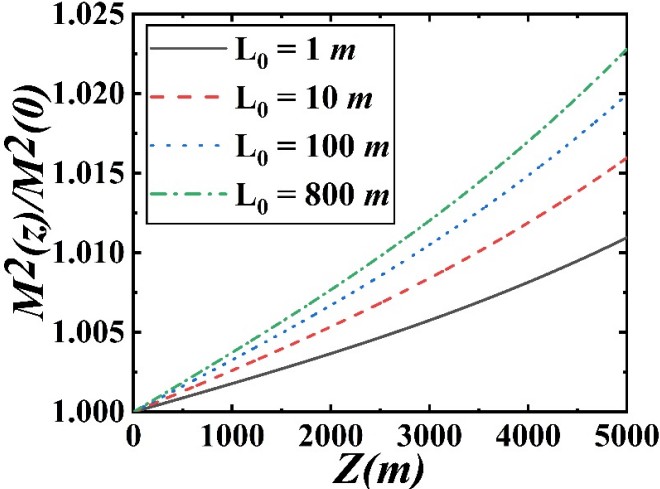

**Figure 3.** Dimensionless quantity $M^2(z)/M^2(0)$ of a partially coherent Airy beam in non-Kolmogorov turbulence versus propagation distance for different $L_0$.

The results shown in Figure 3 excluded the $M^2(z)/M^2(0)$ in four different $L_0$ with $\alpha = 3.9$, $l_0 = 10\,\text{mm}$, $\widetilde{C}_n^2 = 10^{-15}\,\text{m}^{3-\alpha}$, $a = 0.08$, $w_0 = 50\,\text{mm}$, $\sigma_g = 1\,\text{mm}$ and $\lambda = 1064\,\text{nm}$. From Figure 3, it is seen that the $M^2(z)/M^2(0)$ increases by increasing

the outer scale, $L_0$. Obviously, a worse beam quality is obtained by a larger outer scale of non-Kolmogorov turbulence.

We further examined the effect of inner scales on the beam quality in Figure 4. We chose four different values of the inner scales, $l_0$, to be 1, 5, 10 and 20 mm in Figure 4. Clearly, the $M^2(z)/M^2(0)$ increases with a decrease of $l_0$. The reason is that a small $l_0$ means large turbulence intensity.

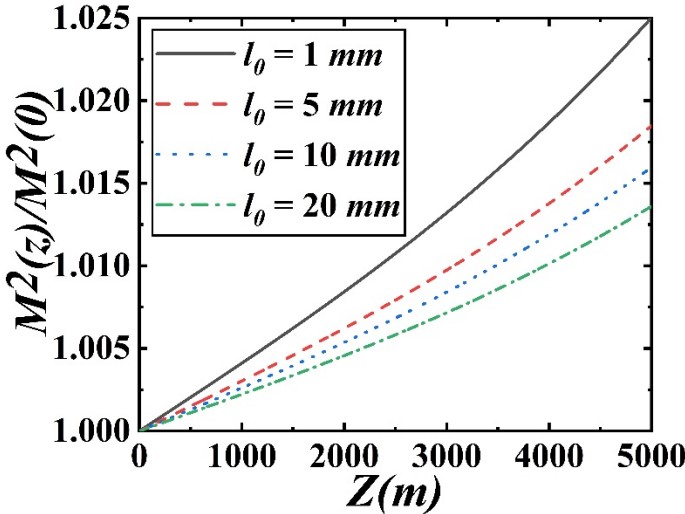

**Figure 4.** Dimensionless quantity $M^2(z)/M^2(0)$ of a partially coherent Airy beam in non-Kolmogorov turbulence versus propagation distance for different $l_0$.

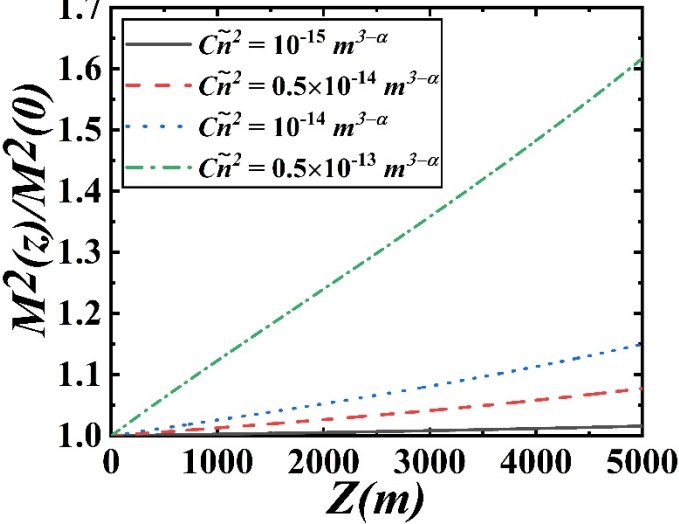

**Figure 5.** Dimensionless quantity $M^2(z)/M^2(0)$ of a partially coherent Airy beam in non-Kolmogorov turbulence versus propagation distance for different $\widetilde{C}_n^2$.

It is generally supposed that the temperature fluctuations of turbulence induce a random behavior in the field of atmospheric index of refraction. For non-Kolmogorov turbulence, the refractive index structure constants are generalized structure constants ($\widetilde{c}_n^2$) with the units of $m^{3-\alpha}$. To further test the relationship of generalized structure constants with the relative beam quality factor, we show the $M^2(z)/M^2(0)$ as a function of the z in Figure 5. The other parameters are $l_0 = 10\,mm$, $L_0 = 10\,m$, $\alpha = 3.9$, $a = 0.08$, $w_0 = 50\,mm$, $\sigma_g = 1\,mm$ and $\lambda = 1064\,nm$. The results of Figure 5 indicate that the relative beam quality decreases by decreasing the generalized structure constants.

The relative beam quality factor with different values of exponential decay factor $a$ is assessed in Figure 6. The other parameters are $\alpha = 3.9$, $l_0 = 10\,\text{mm}$, $L_0 = 10\,\text{m}$, $\widetilde{C}_n^2 = 10^{-15}\,\text{m}^{3-\alpha}$, $w_0 = 50\,\text{mm}$, $\sigma_g = 1\,\text{mm}$ and $\lambda = 1064\,\text{nm}$. It may be an interesting observation in this work that the $M^2(z)/M^2(0)$ has a maximum of $a = 0.63$. In an earlier study, we showed that the Airy beam becomes a Gaussian beam when $a = 0.63$. As discussed above, it was discovered that the Airy beams showed more beneficial influences on beam quality than Gaussian beams.

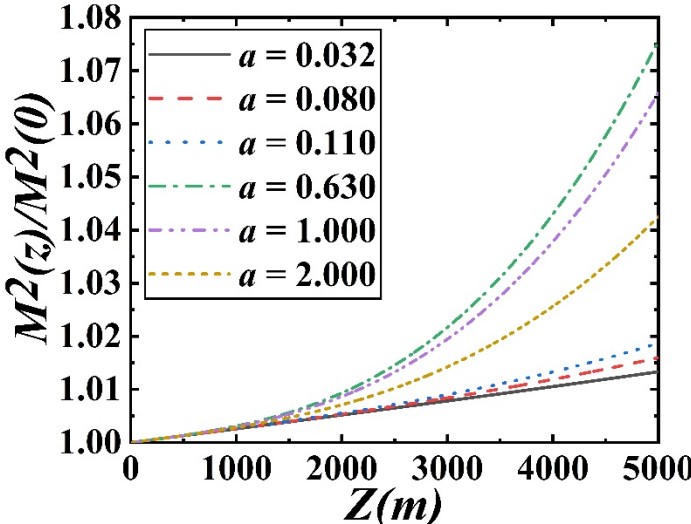

**Figure 6.** Dimensionless quantity $M^2(z)/M^2(0)$ of a partially coherent Airy beam in non-Kolmogorov turbulence versus propagation distance for different $a$.

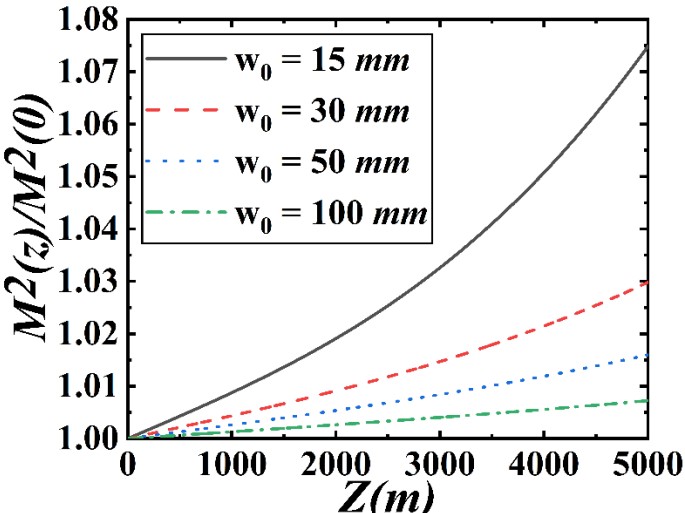

**Figure 7.** Dimensionless quantity $M^2(z)/M^2(0)$ of a partially coherent Airy beam in non-Kolmogorov turbulence versus propagation distance for different $w_0$.

For Airy beams, the scale factor $w_0$ is similar to the Gaussian beam's waist. The scale factor is important to Airy beams because the total power of PCA beams is proportional to $w_0^2$ [47]. To determine whether the scale factor also had an effect on the beam quality, we performed the $M^2(z)/M^2(0)$ for four different $w_0$ with $\alpha = 3.9$, $l_0 = 10\,\text{mm}$, $L_0 = 10\,\text{m}$, $\widetilde{C}_n^2 = 10^{-15}\,\text{m}^{3-\alpha}$, $a = 0.08$, $\sigma_g = 1\,\text{mm}$ and $\lambda = 1064\,\text{nm}$ in Figure 7. As shown in Figure, the quality factor increases by decreasing the scale factor. Accordingly, we consider the possibility that the higher-intensity PCA beams have superiority to PCA beams with lower-intensity in non-Kolmogorov turbulence.

Spatial coherence is the simplest case that can tell us about the phase relationship between the field amplitudes at two points to the laser beam. The existence, or not, of spatial coherence between the fields at two points can be demonstrated in a Young's slits experiment. To investigate the PCA beams' coherence effect on the quality, we show the $M^2(z)/M^2(0)$ against z for four $\sigma_g$ in Figure 8. The calculation parameters are $\alpha = 3.9$, $l_0 = 10$ mm, $L_0 = 10$ m, $\widetilde{C}_n^2 = 10^{-15}$ m$^{3-\alpha}$, $a = 0.08$, $w_0 = 50$ mm and $\lambda = 1064$ nm. It can be concluded that the $M^2(z)/M^2(0)$ also increases in non-Kolmogorov turbulence, but the increments are slower when its coherence length decreases. As demonstrated in Figure 8, similar results were obtained when reducing the coherence of the PCA beam. As discussed above, it was discovered that the PCA beam with lower coherence showed beneficial influences on non-Kolmogorov turbulence. The physical reason behind this phenomenon is that the partially coherent Airy beam consists of multiple modes, and their distortions in non-Kolmogorov turbulence are less correlated.

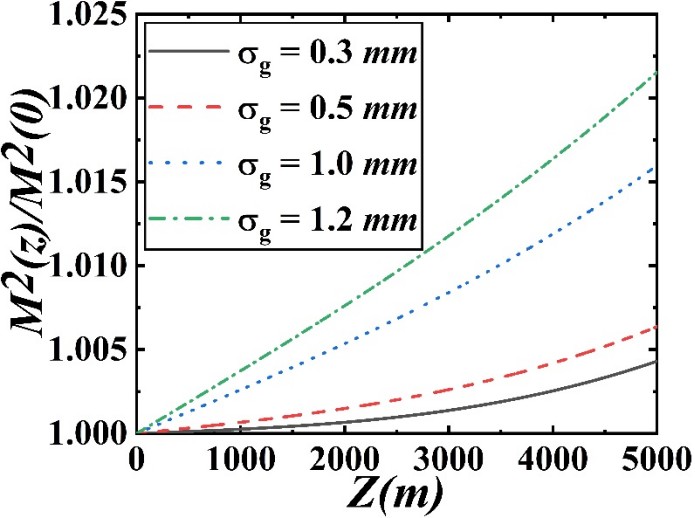

**Figure 8.** Dimensionless quantity $M^2(z)/M^2(0)$ of a partially coherent Airy beam in non-Kolmogorov turbulence versus propagation distance for different $\sigma_g$.

## 4. Summary

The beam quality factor of a PCA beam in non-Kolmogorov turbulence has been studied. The beam quality was reduced effectively for a shorter outer scale, a longer inner scale, a lower generalized structure constant of non-Kolmogorov turbulence, a larger scale factor and a smaller coherent length of the initial beam. The poorest beam quality of a PCA beam is achieved for $\alpha \approx 3.11$ and $a = 0.63$. Our methods can be extended to research the beam quality of other structured laser beams.

**Author Contributions:** Conceptualization, W.W. and X.M.; data curation, W.W. and X.M.; writing—original draft preparation, W.W. and S.C.; writing—review and editing, W.W.; supervision, W.W.; project administration, W.W. and X.M. All authors have read and agreed to the published version of the manuscript.

**Funding:** This work was supported by the Hunan Provincial Natural Science Foundation of China (2020JJ4491). This work was supported in part by the Huaihua University Double First-Class initiative Applied Characteristic Discipline of Control Science and Engineering.

**Institutional Review Board Statement:** Not applicable.

**Informed Consent Statement:** Not applicable.

**Data Availability Statement:** Not applicable.

**Conflicts of Interest:** The authors declare that they have no known competing financial interest or personal relationships that could have appeared to influence the work reported in this paper.

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
