# Peer review of "Beam Quality Factor of Partially Coherent Airy Beam in Non-Kolmogorov Turbulence"

_atmosphere, doi:10.3390/atmos13122061_

Round 1

Reviewer 1 Report

The article shows studies of the beam quality coefficient (?2) for a partially coherent Airy (PCA) beam under conditions of non- Kolmogorov turbulence.

To analyze the quality of the PCA beam, the authors formulated a universal formula of the ?2 coefficient. Based on the universal formula of the ?2 coefficient, numerical studies were done investigating the influence of the various parameters on the quality of the PCA beam in a non-Kolmogorov medium. According to the results of the analysis, it became clear that the worst indicators of the PCA beam could be achieved for a = 0.63 (exponential attenuation coefficient) and ? = 3.11 (generalized parameter of the exponent of non-Kolmogorov turbulence). According to the study of the influence of other parameters, the quality of PCA beam rays improves with: a decrease in ?0 (external turbulence scales), an increase in ?0 (internal turbulence scales), a decrease in ??2 (generalized structural constant of the non-Kolmogorov medium), an increase in ?0 (scale factor), a decrease in ?? (coherence length).

The paper describes well the mathematical part of the formulation of the universal formula of the ?2 coefficient and provides extensive results of numerical modeling with a detailed description. It would also be nice to compare the results obtained with experimental ones. In fact, authors declare the possibility of using their methods to study laser beams of a different structure and thus I think had to present the efficiency of their approach.

Reviewer 2 Report

The authors of this work studied numerically the beam quality factor of a partially coherent Airy beam propagating through a non-Kolmogorov turbulence. The analytic expression for the beam quality factor has been obtained with the help of the Wigner distribution function. The simulation results obtained in this work showed that the beam quality factor can be reduced by controlling the coherence property of the partially coherent Airy beam and the parameters of the non-Kolmogorov turbulence. The findings in this work are important for the free-space optical communications with the partially coherent light beams. 

Overall, the paper is well written and properly documented. The formulae and derivations in the paper look correct. The findings in this work are useful. Therefore, I recommend this work for publication.

Minor revisions:

1. The authors showed that with the decrease of the coherence width of the partially coherent Airy beam, the beam quality factor decreases during propagation. The physical reasons behind this phenomenon can be discussed in the paper.

2. Line 17, “coherent length” should be “coherence length”.

3. The meaning of “WDF” was not included in the paper.

Round 2

Reviewer 1 Report

In general experiments should proof the theoretical work. But, I guess this is not the subject of this paper.